# RVG Peptide-Functionalized Favipiravir Nanoparticle Delivery System Facilitates Antiviral Therapy of Neurotropic Virus Infection in a Mouse Model

**DOI:** 10.3390/ijms24065851

**Published:** 2023-03-19

**Authors:** Meishen Ren, You Zhou, Teng Tu, Dike Jiang, Maonan Pang, Yanwei Li, Yan Luo, Xueping Yao, Zexiao Yang, Yin Wang

**Affiliations:** 1Key Laboratory of Animal Diseases and Human Health of Sichuan Province, Animal Quarantine Laboratory, College of Veterinary Medicine, Sichuan Agricultural University, Chengdu 611130, China; 2Law Sau Fai Institute for Advancing Translational Medicine in Bone and Joint Diseases (TMBJ), School of Chinese Medicine, Hong Kong Baptist University, Hong Kong SAR, China; 3Guangdong-Hong Kong-Macao Greater Bay Area International Research Platform for Aptamer-Based Translational Medicine and Drug Discovery (HKAP), Hong Kong SAR, China; 4Institute of Integrated Bioinformedicine and Translational Science (IBTS), School of Chinese Medicine, Hong Kong Baptist University, Hong Kong SAR, China

**Keywords:** antiviral therapy of neurotropic virus infection, drug CNS delivery, favipiravir

## Abstract

Neurotropic viruses severely damage the central nervous system (CNS) and human health. Common neurotropic viruses include rabies virus (RABV), Zika virus, and poliovirus. When treating neurotropic virus infection, obstruction of the blood–brain barrier (BBB) reduces the efficiency of drug delivery to the CNS. An efficient intracerebral delivery system can significantly increase intracerebral delivery efficiency and facilitate antiviral therapy. In this study, a rabies virus glycopeptide (RVG) functionalized mesoporous silica nanoparticle (MSN) packaging favipiravir (T-705) was developed to generate T-705@MSN-RVG. It was further evaluated for drug delivery and antiviral treatment in a VSV-infected mouse model. The RVG, a polypeptide consisting of 29 amino acids, was conjugated on the nanoparticle to enhance CNS delivery. The T-705@MSN-RVG caused a significant decrease in virus titers and virus proliferation without inducing substantial cell damage in vitro. By releasing T-705, the nanoparticle promoted viral inhibition in the brain during infection. At 21 days post-infection (dpi), a significantly enhanced survival ratio (77%) was observed in the group inoculated with nanoparticle compared with the non-treated group (23%). The viral RNA levels were also decreased in the therapy group at 4 and 6 dpi compared with that of the control group. The T-705@MSN-RVG could be considered a promising system for CNS delivery for treating neurotropic virus infection.

## 1. Introduction

The pathogenic process of neurotropic virus infection and replication cause severe damage to neuronal cells, astrocytes, or microglia in the brain, leading to severe brain edema, encephalitis, and myelitis [1,2,3]. Common neurotropic viruses include poliovirus, vesicular stomatitis virus (VSV), Zika virus (ZIKV), and rabies virus (RABV). More recently, severe acute respiratory syndrome coronavirus (SARS-CoV-2) was also reported to exhibit neurotropism in infected patients [4,5,6]. These viral diseases seriously endangered animal health for a long time, causing substantial economic losses. When antiviral drugs are used to treat neurotropic virus infections, the blood-brain barrier blocks drug exchange between peripheral blood and CNS, thereby inhibiting therapeutic efficacy [7]. For the prevention of neurotropic virus infection, it is challenging for neutralizing antibodies produced by virus vaccination to enter the CNS through the BBB [8,9,10].

As a broad-spectrum inhibitor of RNA-dependent RNA polymerase (RdRp) in RNA viruses, favipiravir (T-705) significantly inhibited viral load in early studies [11,12,13]. According to antiviral studies in the cellular models, T-705 was also reported to inhibit virus replications of ZIKV, RABV, and SARS-CoV-2 [14,15,16]. However, high-dose drugs have to be continuously administered via intranasal (i.n.) or intravenous (i.v.) inoculation to maintain drug concentrations in the brain [17]. Moreover, high-dose administration can cause serious side effects to patients. For ideal antiviral treatment, efficient delivery of drugs to the CNS can promote reduced drug dosage and increased drug concentrations in the brain, thus contributing to therapeutic efficacy. Therefore, a drug encapsulation system with intracerebral delivery efficacy is essential for the antiviral treatment of neurotropic viral diseases.

Rabies virus glycopeptide (RVG) is a polypeptide fragment consisting of 29 amino acids in the rabies virus G protein [18]. As a specific ligand, the RVG peptide can recognize the γ-aminobutyric acid (GABA) and the nicotinic acetylcholine receptor (nAChR) to promote viral transportation to the CNS [7,19,20]. The RVG peptide can be modified in the drug delivery system to improve its intracerebral delivery efficiency to facilitate the study of neuroscience issues in the brain.

VSV virus is a typical neurotropic virus with a single-stranded negative-sense RNA genome and a member of the *Rhabdoviridae* family [21]. In mammals, VSV infection results in vesicular lesions, papules, and erosions with neurotropic symptoms [21]. The infection of VSV is lethal to animals when inoculated into the brain. Its natural infection to human bodies is less toxic, which promotes the development of VSV genome-based reverse genetic vector in the study of vaccines and transsynaptic labeling of CNS neurons [22,23,24,25]. The neurotropism of VSV has been widely reported in previous studies [5,26]. For antiviral studies against neurotropic virus infection, VSV is a suitable model for studying viral infection in the CNS. Herein a T-705@MSN-RVG nanoparticle was constructed and validated in a VSV-infected mouse model. This nanoparticle could recover approximately 77% of mice, which provides a promising strategy for intracerebral drug delivery and antiviral research in the CNS.

## 2. Results

### 2.1. Analysis of Particle Size and Zeta Potential

According to the synthesis steps, the particle size can be progressively increased after drug packaging and peptide modification. The transmission electron microscope (TEM) data of the products from the synthesis steps were detected and statistically analyzed using the Image J and OriginPro 2018C software. The particle sizes of MSN, T-705@MSN, and T-705@MSN-RVG were 120.6 nm ± 10.93 (Figure 1A), 167.7 nm ± 17.91 (Figure 1B), and 238.2 nm ± 14.6 (Figure 1C), respectively.

The changes in surface zeta potential reflected surface charge differences at the synthesis steps. The MSN was negatively charged due to the carboxyl group on its surface, exhibiting a zeta potential of −16.27 mV ± 1.98 (Figure 1D). After packaging T-705 in the MSN, the surface zeta potential was −9.048 mV ± 5.676 (Figure 1E). For the final synthesis product, T-705@MSN-RVG exhibited a zeta potential of −26.7 mV ± 1.61 (Figure 1F) after being conjugated with RVG peptide on its surface.

### 2.2. Characterization and Release Profile Analysis

The ultraviolet (UV) spectrums of RVG and T-705@MSN-RVG were determined to investigate the absorption peak at around 530 to 570 nm. This peak represented the Cy3 fluorescence on the RVG modified on the surface of the nanoparticle. The result indicated that the peak exhibited at around 530 to 570 nm of both RVG and T-705@MSN-RVG. (Figure 2A). A positive correlation was also observed between fluorescence intensity and the concentration gradients of T-705@MSN-RVG (Figure 2B). These results indicated that the RVG was successfully conjugated on the nanoparticle.

The standard solution was first prepared for quantitative detection of T-705. The favipiravir was serially diluted at concentrations of 40, 30, 20, 10, and 5 μg/mL. The diluents of each concentration were detected by the high-performance liquid chromatograph (HPLC) to obtain the values of peak area which represented the drug amount in samples. The obtained values of peak areas and the known concentrations (Appendix A) were used to plot the standard curve and its equation (Figure 2C). For the final product, T-705@MSN-RVG, its encapsulation rate (61.63%) is calculated by dividing the total encapsulated drug amount (8.13 mg) by the total amount of drug input (13.2 mg) before the package in the progress of nanoparticle synthesis. The drug loading rate (13.9%) was obtained by using the total encapsulation (8.13 mg) to divide the sum of the total encapsulation (8.13 mg) and the input MSN (50 mg). For drug delivery systems, the efficacy of sustainably releasing needs to be examined. The accumulative released concentrations of the samples were calculated by using their peak area values and the standard equation. It was indicated that the cumulative drug release gradually increased from 4.44 μg/mL at 3 h to 17.46 μg/mL at 192 h (Figure 2D). The cumulative drug release rate increased from 7.99% at 3 h to 31.4% at 192 h (Figure 2D).

### 2.3. TEM Analysis and Elemental Scanning of T-705@MSN-RVG

As described in the results of the particle size analysis, the diameter of the nanoparticles became larger after drug encapsulation and RVG modification under TEM visualization (Figure 2E). The results of the elemental scanning assay can promote analyzing the changes in the elemental distribution of nanoparticles after drug encapsulation and RVG modifications. As shown in Figure 2(F(b)), the abundant distribution of carbon elements on the MSN surface represents the RVG peptide modification. The carbon elements inside the nanoparticles are indicative of the encapsulated T-705 (Figure 2(F(b))). The oxygen and silicon elements representing MSN were uniformly distributed on the spheres (Figure 2(F(c),F(d))).

### 2.4. Cytotoxicity Analysis of T-705 and T-705@MSN-RVG

In order to evaluate the cytotoxicity, the mouse neuroblastoma (N2a) cells were inoculated with T-705 or T-705@MSN-RVG and harvested for cell viability test. No significant difference in cell viability was observed for T-705 treated cells at concentrations lower than 3 μg/mL (Figure 3A). Since the T-705 was dissolved in the DMSO-containing solution, the cell cytotoxicity caused by this solution was measured as well. In the results, the DMSO did not cause significant cytotoxicity to cells at concentrations lower than 2.9 μL/mL^−1^ (Figure 3B). The concentration of DMSO in each well exhibited in Figure 3A was 0.1 μL/mL^−1^, which did not exceed the safe concentration determined in Figure 3B (2.9 μL/mL^−1^). Simultaneously, the cytotoxicity of T-705@MSN-RVG was determined in cells. According to the results, no observed difference in cell viability at nanoparticle concentrations lower than 2 mg/mL (Figure 3C). According to this cytotoxicity analysis, the maximum safe doses of T-705 and T-705@MSN-RVG were 3 μg/mL and 2 mg/mL, respectively.

### 2.5. Virus Inhibition by T-705 in VSV-Infected Cells

The virus titers were firstly determined at different hours post-inoculation to obtain the growth curves of VSV-GFP at the multiplicities of infection (MOI) of 0.01 and 1, respectively (Figure 4A). In the cellular infection model, the VSV-GFP infected (MOI = 0.01) N2a cells were treated with gradient dilutions of T-705 to investigate the virus inhibition effect. It was indicated that the virus titers were significantly reduced when the concentration of T-705 reached up to 0.75 μg/mL for VSV-GFP (Figure 4B). The virus titer reduced from about 6 Log FFU/mL to 1.5 Log FFU/mL after being treated with T-705 at 3 μg/mL. Moreover, the viral spread between VSV-infected and neighboring cells was analyzed after the cells were treated with 100 μL of T-705 (3 μg/mL) for 48 h. Since the cells were fixed by agarose, the viral particles cannot be released into the supernatant for rapid diffusion as in the liquid culture model. Instead, virus diffusion was completed from infected cells to their neighbor cells. The results showed a minor fluorescent spot in T-705 treated cells rather than in non-treated cells (Figure 4C). Thus, the viral spread was also inhibited by T-705 in vitro.

### 2.6. Broad-Spectrum Analysis of T-705 against Different RNA Viruses

The lysine site at the motif F of the RNA-dependent RNA polymerase (RdRp) is conservative in different RNA viruses, including positive-stranded RNA viruses and negative-stranded RNA viruses. Since mutation of this lysine at the motif F can attenuate the entry efficiency of T-705 into the RdRp during the inhibition process of viral replication [27,28]. Sequence conservation at this site can contribute to broad-spectrum inhibition against different RNA viruses, especially those with neurotropism. Thus, the amino acid sequences of RdRp were aligned by the software MEME Suite 5.5.1 to access the broad-spectrum properties of T-705. It was indicated that this lysine site is conservative among positive- and negative-stranded viruses, especially those neurotropic viruses (Japanese encephalitis virus, Zika virus, Poliovirus, Coxsackievirus, Vesicular Stomatitis Virus, and Rabies virus) (Figure 4D).

### 2.7. The Distribution of VSV and T-705@MSN-RVG In Vitro and In Vivo

Since the nAChR expressed on the N2a cells can be recognized by the RVG peptide, the RVG can promote cellular uptake of T-705@MSN-RVG in N2a cells. The N2a cells were infected with VSV-GFP at MOI of 1. The infected cells were incubated with T-705@MSN-RVG and stained by Hoechst. It was indicated in the results that the VSV-GFP (green fluorescence signal) and T-705@MSN-RVG (red fluorescence signal) exhibited co-location in the N2a cellular model (Figure 5A).

In the mouse infection model, fluorescence detections in brain sections were critical to investigate the CNS delivery of T-705@MSN-RVG and its distribution with VSV-GFP in mouse brains. Six-week-old female Balb/c mice were infected with VSV-GFP at 200 FFU by i.c. route and 25 μL of T-705@MSN-RVG (1 μg/mL) by i.v. injection. The brain sections were prepared 48 h post-infection to be detected under a fluorescence microscope. In the results, Cy3 red fluorescence signal can be detected in the section prepared from the mouse inoculated via i.v. route, indicating that the T-705@MSN-RVG can be delivered into the mouse brain (Figure 5B). Moreover, the GFP and Cy3 overlapped, indicating that the T-705@MSN-RVG have the same cellular distribution as VSV-GFP in the mouse brain (Figure 5B).

### 2.8. Antiviral Experiments in Mouse Brains by Using T-705@MSN-RVG

For the in vivo experiments, the appropriate viral injection dose was first determined by pathogenicity experiments. Nearly 80% of the mice were sacrificed at 21 dpi with 100 FFU of VSV-GFP injection, whereas 100% of the mice were sacrificed under 200 FFU (Figure 6A). Thus, 100 FFU of virus inoculation was used to construct the mouse infection model in an in vivo antiviral experiment. For intracerebral antiviral experiments, the efficiency of intracerebral drug delivery severely influences the therapeutic effect. The unpackaged T-705 was first investigated by i.v. inoculation in the mouse infection model (100 FFU). The results showed no difference in the percent survival rate between the VSV virus control group and the VSV+T-705 treatment group (Figure 6B). Subsequently, the antiviral effect of T-705@MSN-RVG was evaluated in the mouse infection model. As shown in the diagram (Figure 6C), the mice were infected with VSV-GFP by i.c. route and subsequently inoculated with T-705@MSN-RVG via i.v. route. At 21 dpi, the percentage weight changes and percent survival of 3 groups of mice were analyzed. Moreover, the intensity ratios of intracerebral CD45-positive cells and the vRNA copy numbers were determined at 2, 4, and 6 dpi (Figure 6C). The body weight in the VSV virus control group was 80–85% compared to that of the T-705@MSN-RVG drug control group during 2 to 5 dpi (Figure 6D). Whereas no noticeable difference in weight change was found in the VSV+T-705@MSN-RVG treatment group (Figure 6D). According to the results, a significant difference in percent survival was observed between the VSV virus control group (3 of 13 mice survived; 23% of survival rate) and the VSV+T-705@MSN-RVG treatment group (10 of 13 mice survived; 77% of survival rate) (Figure 6E).

The changes in inflammation and viral load were investigated by detecting the intensities of CD45-positive cells and the vRNA levels in mouse brains at 2, 4, and 6 dpi. According to the results, significant CD45-positive cells were observed in the virus control group (VSV-infected) (Figure 7A). However, no obvious sign was found in the mock-infected (mock-infected T-705@MSN-RVG) and therapy groups (VSV-infected T-705@MSN-RVG) (Figure 7A). The intensities of CD45-positive cells in each sample were quantified to analyze the difference in CD45-positive intensities between the virus control group and therapy group at 2, 4, and 6 dpi. The results showed that the CD45 intensity level of the therapy group was less than that of virus control group at 2, 4, and 6 dpi, respectively (Figure 7B). In the mouse brain, no significant difference in vRNA copy number was found at 2 dpi. The vRNA copy numbers of the therapy group were separately reduced at 4 and 6 dpi compared with that of the virus control group (Figure 7C).

## 3. Discussion

For treating neurotropic virus infections, obstruction of the BBB leads to a reduction in drug delivery efficiency to the CNS. As a result, the drug is gradually consumed and metabolized in the peripheral blood and organs. In turn, the practical drug components are not delivered to the target cells, and the therapeutic efficacy is thus limited. In previous studies, available treatment of neurotropic virus infections required high doses of drugs administered, causing severe adverse effects [14,29,30]. Therefore, drug delivery to the CNS is essential to improving the therapeutic effect on neurotropic virus infection.

For the characterization of T-705@MSN-RVG, the elemental mapping assay was performed to analyze the element distribution on this nanoparticle. It was indicated that the oxygen and silicon were evenly concentrated into a circle to form the MSN (Figure 2(F(c),F(d))). The element carbon was introduced in this system when MSN was loaded with the T-705 and modified with the RVG. The outer layer of carbon indicated the location of the RVG, and the inner layer demonstrated the distribution of the T-705 (Figure 2(F(b))). In addition, both RVG and T-705@MSN-RVG exhibited representative peaks at the wavelength of 530–570 (Figure 2A), which indicated that the RVG was modified on the nanoparticle.

The rabies virus glycopeptide (RVG) was reported to bind the acetylcholine receptor. According to the previous study, the RVG was modified on the surface of a virus-like gold nanorod to provide CNS delivery efficiency [31]. In this study, the mesoporous silicon (MSN) was modified with the RVG to provide brain delivery efficiency. The fluorescent signals of Cy3 and GFP were visualized to analyze the distribution of T-705@MSN-RVG and VSV viral particles in vitro and in vivo, respectively. The results showed significant co-localizations in the N2a cellular model and mouse brain (Figure 5A,B). It could be observed that the co-localization of fluorescent signals (Cy3 and GFP) existed in the cytoplasm. Therefore, it may be inferred that the T-705@MSN-RVG may enter cells through receptor-mediated endocytosis, which was also mentioned in a previous study [32].

In addition, nanoparticle endocytosis could promote intracellular drug release as well. For favipiravir (T-705) or other RNA-dependent RNA polymerase inhibitors, intracellular drug release is essential for inhibiting RNA virus replication in the cytoplasm. After being released in the cytoplasm, the T-705 monomer is sustained by phosphoribosylation and phosphorylation to form the activated favipiravir-RTP (F-RTP) [33]. The activated F-RTP can be incorporated into the nascent RNA through the NTP entry pocket on the RdRp during viral transcription and replication in the cytoplasm. Since the conservativeness at this entry pocket on the RdRp was related to the virus inhibition by T-705 [27]. Mutant at the lysine of the motif in the pocket could limit the entry of T-705 in the RdRp, thus attenuating the effect of T-705. Therefore, the motif sequence on the RdRp of positive- and negative-stranded neurotropic RNA viruses was aligned to analyze the conserved lysine. According to the result, the conserved sites were observed in the RdRp of different viruses. It may be inferred that the T-705@MSN-RVG could be available for inhibiting different neurotropic RNA viruses.

For the viral particles in extracellular fluid, the genomic RNA and RdRp were assembled in the virions without being interfered with F-RTP. Therefore, it is vital for antiviral treatment to sustain the drug release in cells. According to the results of this study, the T-705 was released continuously for 192 h at cumulative drug concentrations up to 17.46 μg/mL (Figure 2D). It was indicated that the virus RNA copy numbers were reduced at 4 and 6 dpi after mice were inoculated with T-705@MSN-RVG compared with virus control group. Thus, continuous drug release should be essential to inhibit the vRNA level in the mouse brain.

The antiviral effects of in vivo and in vitro tests are the two most critical indicators for investigating the T-705@MSN-RVG. In the VSV viral infection cellular model, approximately 10,000-fold decreases in virus titers were observed in vitro (Figure 4B). In vivo experiments showed a significantly enhanced survival ratio (77%) after mice were treated with the T-705@MSN-RVG. Thus, it was proved that T-705@MSN-RVG could inhibit virus infection in vitro and in vivo. In addition, this nanoparticle was indicated to inhibit the virus proliferation from the infected cells to their nearby cells in vitro (Figure 4C). Safety evaluation is also a critical section to investigate the nanoparticle. In this study, the effective viral inhibition required for drug concentration was less than the maximum safe concentration without cytotoxicity in vitro and in vivo. In addition, no significant cellular inflammation was found in mice inoculated with T-705@MSN-RVG compared with the virus control group (Figure 7A,B).

## 4. Materials and Methods

### 4.1. Cell Lines, Antibodies, Viruses, and Animals

Murine neuroblastoma (N2a) cells were cultured in Dulbecco’s modified Eagle’s medium (DMEM) containing 12% fetal bovine serum (FBS) (Gibco, New York, NY, USA). BSR (a cell line derived from BHK-21) was cultured in DMEM containing 10% FBS.

Caspase-3 (Invitrogen, Shanghai, China) and CD45 (Abcam, Cambridge, UK) rabbit monoclonal antibodies were stored in our lab. HRP-conjugated goat anti-rabbit antibodies were obtained from Boster (Boster, Wuhan, China).

The VSV-GFP recombinant virus was gifted by Professor Ling Zhao (Huazhong Agricultural University, Wuhan, China). In the mouse experiments, six-week-old female Balb/c mice were purchased from Sichuan Provincial Laboratory Animal Public Service Center, Chengdu, China.

### 4.2. Preparation of T-705@MSN-RVG

The synthesis of the T-705@MSN-RVG nanoparticle was performed according to the following steps. Firstly, 13.2 mg of T-705 powder was dissolved in 660 μL of anhydrous ethanol (containing 10 μL of DMSO). Then, 2 mL of 2.5% mesoporous silicon (MSN) solution was dispersed in anhydrous ethanol and centrifuged at 10,000 rpm for 10 min to remove the excessed ethanol solution. Next, the T-705 solution obtained above was mixed with the MSN precipitation, and the mixture was sonicated in a water bath. After being incubated at 300 rpm overnight, the solution was centrifuged at 10,000 rpm for 10 min to remove the supernatant. The precipitation product was washed by centrifugation three times with 0.02 mM MES buffer (pH 5.5). In the next step, 600 μL of RVG (5 μg/μL) was prepared and mixed with 250 μL of EDC (10 mg/mL) and the T-705@MSN solution obtained in the previous step. The mixture was incubated overnight and centrifuged at 10,000 rpm for 10 min to remove the free RVG. The final product was dispersed in pure water at a concentration of 10 mg/mL.

### 4.3. Characterization of T-705@MSN-RVG

The drug concentration in the T-705@MSN-RVG nanoparticle was determined by an LC20A (SHIMADZU, Kyoto, Japan) high performance liquid chromatograph (HPLC). Briefly, the favipiravir was serially diluted at concentrations of 40, 30, 20, 10, and 5 μg/mL. The diluents were detected by HPLC to obtain the peak area at each concentration. The standard curve of T-705 was plotted using the peak areas as the vertical coordinate and the drug concentration as the horizontal coordinate. The total encapsulated drug was the difference between the amount of drug before packaging and the amount of drug in the supernatant after packaging. The encapsulation rate was calculated by dividing the encapsulation and the amount of drug before packaging. The drug loading rate was obtained by using the encapsulation rate to divide the sum of the total encapsulation and the input MSN. In the cumulative drug release test, 2 mg of T-705@MSN-RVG was diluted in 5 mL of release buffer (1 × PBS solution containing 10% ethanol and 5% Tween-80) in a dialysis bag. The release buffer was collected and detected at 3, 6, 12, 24, 48, 96, and 192 h, respectively, to obtain the cumulative drug release curve.

The UV spectrums of RVG and T-705@MSN-RVG were detected within the wavelength ranged from 200 to 650 nm by a SpectraMax i3x (Molecular Devices, San Jose, CA, USA) microplate reader. The fluorescence intensities of serially diluted T-705@MSN and T-705@MSN-RVG were detected at concentrations of 1.95 to 2000 μg/mL. The main synthesis products were visualized under an HT7800 (HITACHI, Tokyo, Japan) transmission electron microscope (TEM) to determine the quality of the synthesized nanoparticles of MSN and T-705@MSN-RVG. The surface zeta potential of the 3 representative steps was measured on a Malvern Zetasizer Nano Series (Malvern Panalytical, Malvern, UK). The distribution of C, O, and Si elements on the T-705@MSN-RVG was measured by high-angle, annular dark-field (HAADF) in STEM-EDS element mapping system (Bruker Nano GmbH, Berlin, Germany).

### 4.4. Cell Viability Test

The determination of cell viability was performed using the Vybrant MTT Cell Proliferation Assay Kit (Sigma, New York, NY, USA). Briefly, the N2a cells were cultured overnight at 37 °C followed by washing three times with PBS and refreshing the culture medium with DMEM (containing 2% FBS). Subsequently, the T-705 compound was serially diluted in DMEM (containing 2% FBS and DMSO at 0.1 μL/mL^−1^) at final concentrations from 0.125 to 4 μg/mL. The T-705@MSN-RVG was serially diluted in DMEM (containing 2% FBS) at final concentrations from 0.031 to 4 mg/mL. The diluted fractions of T-705 and T-705@MSN-RVG were separately supplied in the cells for 48 h at 37 °C. After being washed three times with PBS, the supernatant of each well was replaced with 100 μL of fresh culture medium (containing 10 μL of 12 mM MTT stock solution). The absorbance at 450 nm was detected after 4 h of incubation at 37 °C. The absorbance at 450 nm ratios of samples to DMEM control were calculated and statistically analyzed.

### 4.5. Virus Infection and Titration

For viral infection, the N2a cells were infected with VSV and RABV at a multiplicity of infection (MOI) of 0.01 or 1 for 1 h at 37 °C. Subsequently, the cells were washed three times with PBS and maintained in DMEM supplemented with 2% FBS for 12, 24, 36, and 48 h at 37 °C. The virus titration was performed according to the previous study [34]. Briefly, the viral supernatant was diluted in a 10-fold gradient (from 10^−1^ to 10^−7^ dilutions) to be added in quadruplicate to 96-well cell culture plates at 100 μL, followed by supplying 100 μL BSR cells (1 × 10^4^ cells/mL) in each well. After 48 h of incubation, the cells were visualized by a TS100 fluorescence microscope (Nikon, Tokyo, JPN).

### 4.6. Virus Inhibition by T-705 In Vitro

To investigate the virus inhibition effect, virus titers and fluorescent spots were determined in T-705 treated cells. The N2a cells were seeded in 24-well cell culture plates at 5 × 10^4^ cells/mL overnight. Then, cells were infected with VSV-GFP at MOI of 0.01 at 37 °C for 1 h. After being washed, cells were inoculated with gradient dilutions of T-705 at concentrations of 0.094 to 3 μg/mL at 37 °C for 48 h. The supernatant was collected and determined by virus titration to analyze the differences in virus titers post T-705 treatment.

The changes in fluorescent spots reflect the viral spread efficacy between cells. Similarly, N2a cells were seeded in 12-well cell culture plates at 1×10^5^ cells/mL overnight. Cells were infected with VSV-GFP at MOI of 0.01 at 37 °C for 1 h. Subsequently, cells were washed and fixed in a medium containing 1 × DMEM, 0.7% low-melting-point agarose, 2%FBS, and T-705 at 3 μg/mL at 37 °C for 48 h. Then, the agarose was removed, and the cells were stained by a DAPI reagent (Thermo Fisher, Waltham, MA, USA) to be visualized by a fluorescence microscope.

### 4.7. Fluorescent Co-Localization of Virus and Nanoparticle

The VSV-GFP is a recombinant virus expressing the green fluorescent protein. The Cy3 fluorescein was modified on the RVG to provide a red fluorescent signal. In the cellular model, the virus-infected N2a cells (MOI of 1) were incubated with 100 μL of T-705@MSN-RVG (2 mg/mL) at 37 °C for 1h. Then, cells were washed three times and stained by a Hoechst (Thermo Fisher, Waltham, MA, USA) at 37 °C for 5 min. Subsequently, the cells were visualized by a TS100 fluorescence microscope (Nikon, Tokyo, Japan).

In the mouse infection model, co-localization was determined in the mouse brain. Briefly, six-week-old female Balb/c mice were infected with VSV-GFP at 200 FFU or DMEM by intracranial (i.c.) route. Then, the mice were inoculated with 25 μL of T-705@MSN-RVG (1 mg/mL) by i.v. injection. The mouse brains were collected at 2 dpi and fixed in 4% paraformaldehyde for 48 h, followed by being dehydrated in 30% sucrose solution and embedded in SAKURA Tissue-Tek^®^ O.C.T compound (SAKURA, Torrance, CA, USA) for rapid freezing and slicing [35]. The slices were stained by Hoechst (Thermo Fisher, Waltham, MA, USA) at 37 °C for 5 min and then visualized by a TS100 fluorescence microscope (Nikon, Tokyo, Japan).

### 4.8. The Antiviral Experiments in the Mouse Infection Model

For the pathogenicity investigation of VSV, six-week-old Balb/c mice in 4 groups were respectively inoculated with VSV-GFP at 200 FFU, 100 FFU, 50 FFU, and DMEM by i.c. route. The mortalities of all the 4 groups (*n* = 10) were recorded for 21 days to investigate the survival rates. The antiviral effect of the T-705 compound was first determined. Six-week-old Balb/c mice in 3 groups (VSV virus-infected group, VSV+T-705 treatment group, and T-705 control group) were infected with VSV-GFP at 100 FFU or DMEM by i.c. route. The 3 groups of mice were inoculated with 25 μL of T-705 compound (3 μg/mL) via i.v. injection (*n* = 10). The mortalities were recorded for 21 days to investigate the survival rates.

For the antiviral experiments in vivo, six-week-old Balb/c mice in 3 groups (VSV virus-infected group, VSV+T-705@MSN-RVG treatment group, and T-705@MSN-RVG control group) were infected with VSV-GFP at 100 FFU or DMEM by i.c. route. Then, these 3 groups of mice were inoculated with 25 μL of T-705@MSN-RVG (2 mg/mL) via i.v. injection. Bodyweight and mortality of all the 3 groups (*n* = 13) were recorded for 21 days to investigate the percentage weight change and survival rate. In addition, mouse brains were collected at 2, 4, and 6 dpi from the same mouse infection and treatment model to determine the changes of CD45 positive cells and virus genomic RNA copies in mice treated by T-705@MSN-RVG.

### 4.9. Quantitative Real-Time PCR (qPCR)

The virus genomic RNA was quantified by qPCR. For cDNA preparation, the mouse brains (*n* = 5) in the antiviral experiments were collected at 2, 4, and 6 dpi. All the brains were homogenized by a JX-FSTPRP-64 homogenizer (Jingxin, Yanbian, China) and centrifuged at 10,000 rpm for 5 min. The supernatant was extracted by TRIZol reagent (Invitrogen, Shanghai, China) and was transcribed into cDNA by a HiScript Reverse Transcriptase kit (Vazyme, Shanghai, China). For qPCR detection, a primer pair set (forward primer 5′-AGTCTAGCTTCCAGCTTCTGA-3′, reverse primer 5′-ACAGGATATTAGTTGTTCGAAAGGC-3′) was designed to detect the genomic RNA of VSV-GFP. The plasmid containing virus genomic RNA was 10-fold gradient diluted and amplified to construct a standard curve. The copy numbers of viral genomic RNA (copies/μg total RNA) in mouse brains were detected by qPCR and calculated based on the standard curve.

### 4.10. Immunohistochemistry of Mouse Brains

In antiviral experiments in vivo, the CD45 and caspase-3 positive cells in the mouse brain were visualized and statistically analyzed to assess cell inflammation and cell apoptosis level. Briefly, 3 groups of six-week-old Balb/c mice were inoculated with VSV-GFP at 100 FFU or DMEM by i.c. injection. Then, the virus-infected groups of mice were injected with 25 μL of T-705@MSN-RVG (2 mg/mL) via i.v. route. The mouse brains were collected at 6 dpi and fixed in 4% paraformaldehyde for 48 h. The fixed brains were embedded in the paraffin wax and cut into 5-μm sections along the sagittal plane. The sections were stained by CD45 and caspase-3 rabbit monoclonal antibodies when the sections were retrieved in a citric acid buffer. All the sections were visualized and scanned under a VS120-S6-W slide scanning system (Olympus, Tokyo, Japan). The intensity of CD45 and caspase-3 positive cells was calculated and statistically analyzed to assess the cellular inflammation and apoptosis levels in mouse brains.

### 4.11. Statistical Analysis

Statistical analysis was performed using the GraphPad Prism 8 (GraphPad Software, Inc., San Diego, CA, USA), OriginPro 2018C (OriginLab, Ltd., Wellesley Hills, MA, USA), and Image J (National Institutes of Health, Bethesda, MD, USA). The motif sequence alignment was performed by the MEME Suite 5.5.1 (https://meme-suite.org). The unpaired two-tailed Student’s *t*-test was performed to evaluate statistical significance. Kaplan–Meier survival curves were used to determine the difference in survival rates between different groups. Statistically significant differences were represented as * *p* < 0.05, ** *p* < 0.01, *** *p* < 0.001, and **** *p* < 0.0001. No significant difference was represented as ns.

## 5. Conclusions

Effective antiviral therapy for neurotropic virus infection is still a fundamental challenge due to the obstruction of the blood-brain barrier. Functionalized nanoparticles are promising candidates for promoting drug CNS delivery. The T-705 is an RdRp inhibitor against various neurotropic RNA viruses. To facilitate antiviral therapy of neurotropic virus infection, a T-705@MSN-RVG nanoparticle packaging the T-705 was constructed and validated for antiviral treatment in a neurotropic VSV-infected mouse model. The results demonstrated that the T-705@MSN-RVG could be effectively delivered into the mouse brain, exhibiting co-localization with the virus. The survival rate was significantly improved when the infected mice were treated with T-705@MSN-RVG. The nanoparticle developed in this study provided a potential strategy for effective drug delivery and antiviral therapy in patients suffering from neurotropic virus infection.

## Figures and Tables

**Figure 1 ijms-24-05851-f001:**
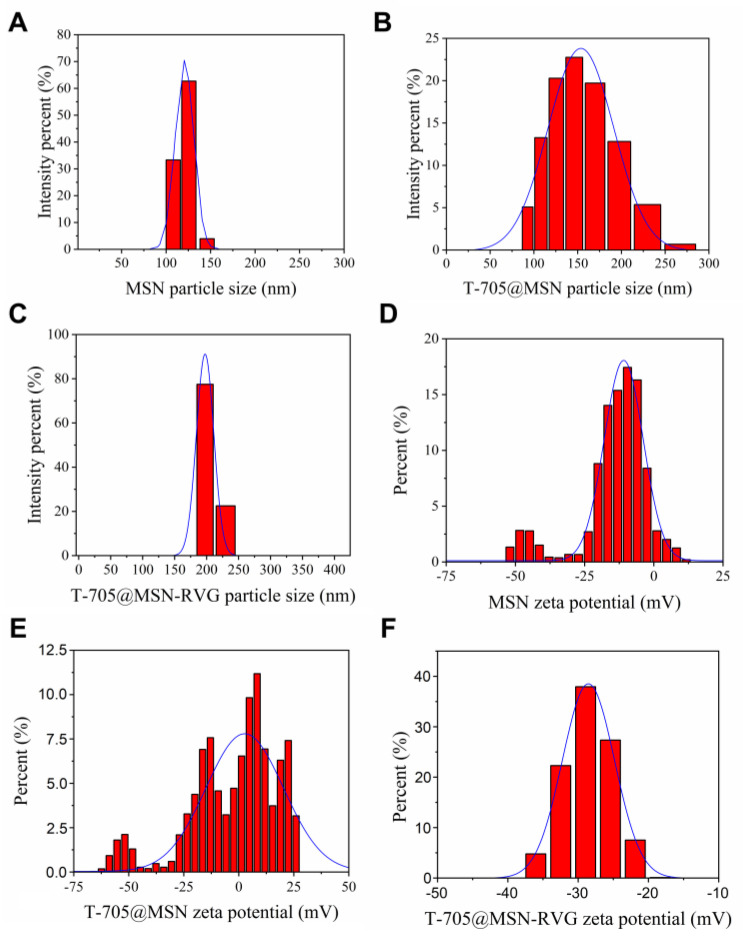
Particle size and zeta potential analysis. The particle size (nm) analysis of (**A**) MSN, (**B**) T-705@MSN, and (**C**) T-705@MSN-RVG. The surface zeta potential (mV) determinations of (**D**) MSN, (**E**) T-705@MSN, and (**F**) T-705@MSN-RVG.

**Figure 2 ijms-24-05851-f002:**
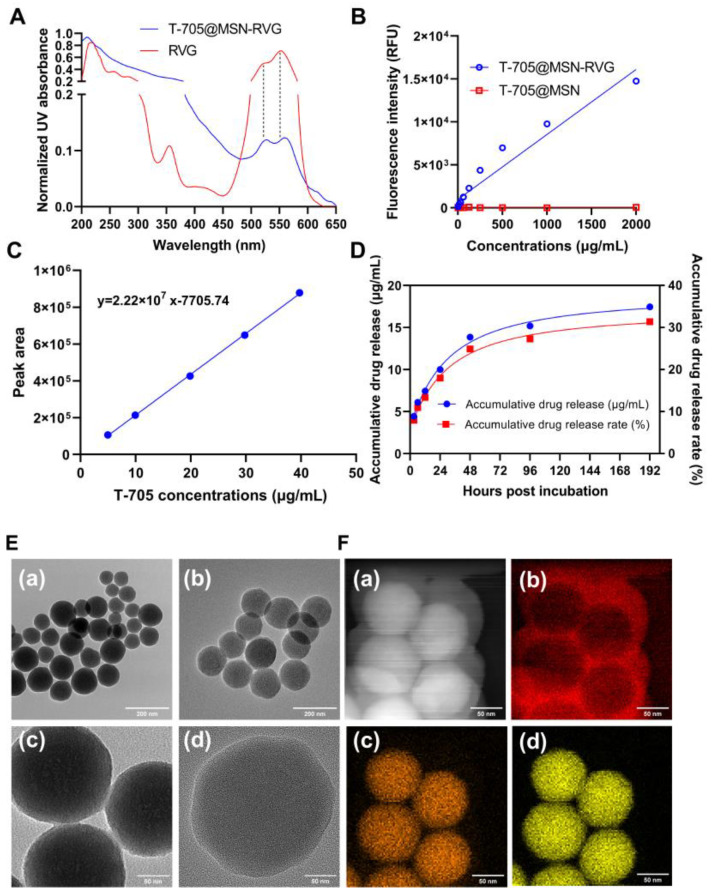
Characterization and drug release profile analysis of T-705@MSN-RVG. (**A**) The ultraviolet (UV) spectrums of RVG and T-705@MSN-RVG were determined at 200 to 650 nm wavelengths. (**B**) The serially diluted T-705@MSN-RVG and T-705@MSN at concentrations of 1.95 to 2000 μg/mL were tested to investigate the fluorescence intensities of Cy3. (**C**) The favipiravir was serially diluted at concentrations of 40, 30, 20, 10, and 5 μg/mL to be detected to calculate the standard curve. (**D**) To investigate the drug release curve, 2 mg of T-705@MSN-RVG was diluted in 5 mL of release buffer to be detected at 3, 6, 12, 24, 48, 96, and 192 h, respectively. (**E**) The TEM images of MSN (panel (**a**), scale bar 200 nm; panel (**c**), scale bar 50 nm) and T-705@MSN-RVG (panel (**b**), scale bar 200 nm; panel (**d**), scale bar 50 nm). (**F**) Elemental mapping of T-705@MSN-RVG, scale bar 50 nm. Panel (**a**) presented the bright field. The elements of carbon, oxygen, and silicon were exhibited in panels (**b**–**d**).

**Figure 3 ijms-24-05851-f003:**
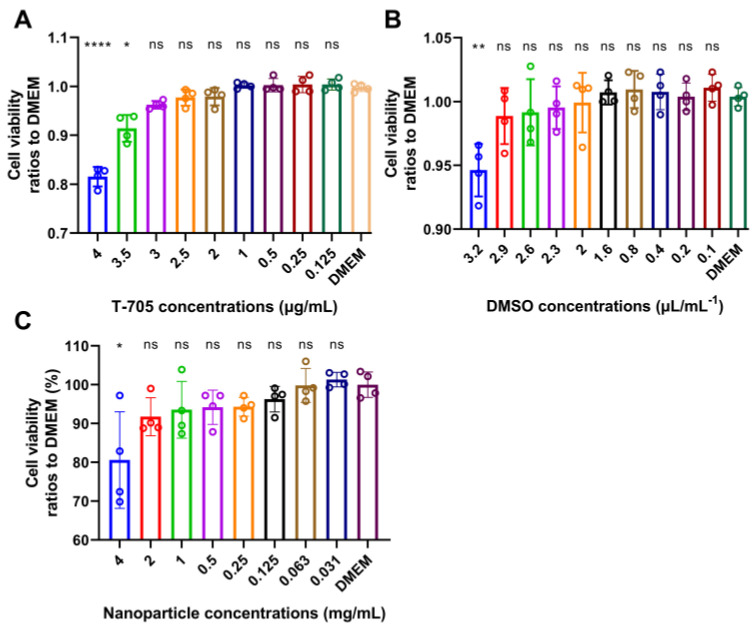
Cytotoxicity analysis of T-705@MSN-RVG. In the cellular model, the cell viability ratios to DMEM were determined after cells were respectively incubated with T-705, DMSO, and T-705@MSN-RVG. (**A**) N2a cells were incubated with gradient-diluted T-705 at concentrations of 0.125 to 4 μg/mL. (**B**) N2a cells were incubated with gradient-diluted DMSO at concentrations of 0.1 to 3.2 μL/mL^−1^. (**C**) N2a cells were incubated with gradient-diluted T-705@MSN-RVG at concentrations of 0.031 to 4 mg/mL. Statistical analysis of grouped comparisons was carried out by Student’s *t*-test (ns represents not significant, * *p* < 0.05; ** *p* < 0.01; **** *p* < 0.0001). The bar graph represents means ± SD, *n* = 4.

**Figure 4 ijms-24-05851-f004:**
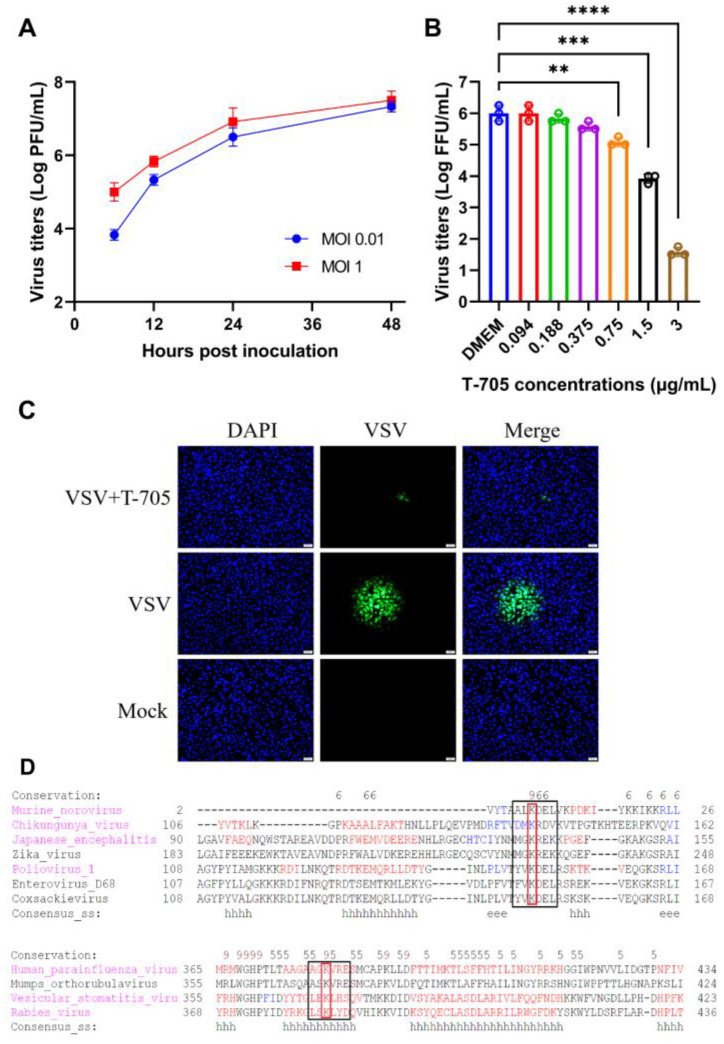
Viral inhibition by T-705 and the motif alignment at the entry site in the RdRp. (**A**) N2a cells were infected with VSV-GFP at MOI of 0.01 and 1. Virus titers in the supernatant were determined at 6, 12, 24, and 48 h post-infection (hpi). The obtained virus titers at different hpi were used to plot the virus growth curves. (**B**) VSV-GFP infected (MOI = 0.01) N2a cells were treated with gradient dilutions of T-705 at concentrations of 0.094 to 3 μg/mL. (**C**) VSV-GFP infected (MOI = 0.01) cells were incubated with 100 μL of T-705 (3 μg/mL) for 48 h. The fluorescent spot was visualized by a fluorescence microscope in T-705 treated group, VSV virus control group, and mock-infected group. Scale bar 20 μm. (**D**) Motif alignment at the entry site of T-705 in the RdRp from positive- and negative-stranded neurotropic RNA viruses. Statistical analysis of grouped comparisons was carried out by Student’s *t*-test (ns represents not significant, ** *p* < 0.01; *** *p* < 0.001; **** *p* < 0.0001). The bar graph represents means ± SD, *n* = 3.

**Figure 5 ijms-24-05851-f005:**
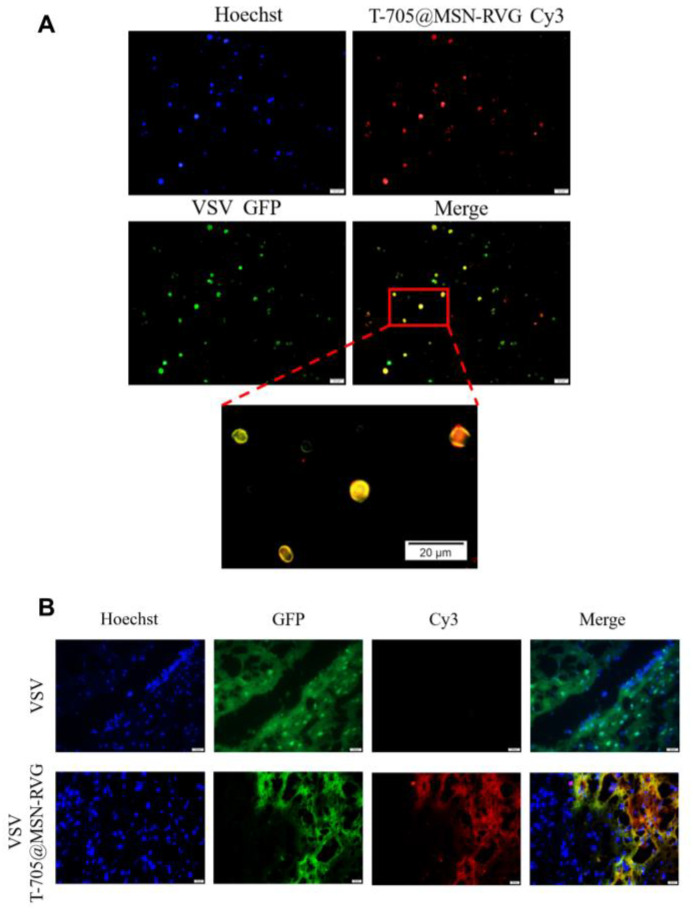
The distribution of VSV and T-705@MSN-RVG in vitro and in vivo. (**A**) N2a cells (VSV-GFP infected at MOI of 1) were incubated with 100 μL of T-705@MSN-RVG (2 mg/mL) at 37 °C for 1 h. After being washed and stained by Hoechst, the cells were visualized by a fluorescence microscope. (**B**) Six-week-old female Balb/c mice were inoculated with VSV-GFP (200 FFU, i.c.) and T-705@MSN-RVG (25 μL at a concentration of 1 mg/mL, i.v.). The fluorescences of Hoechst, GFP, and Cy3 were determined by a fluorescence microscope. Scale bar 20 μm.

**Figure 6 ijms-24-05851-f006:**
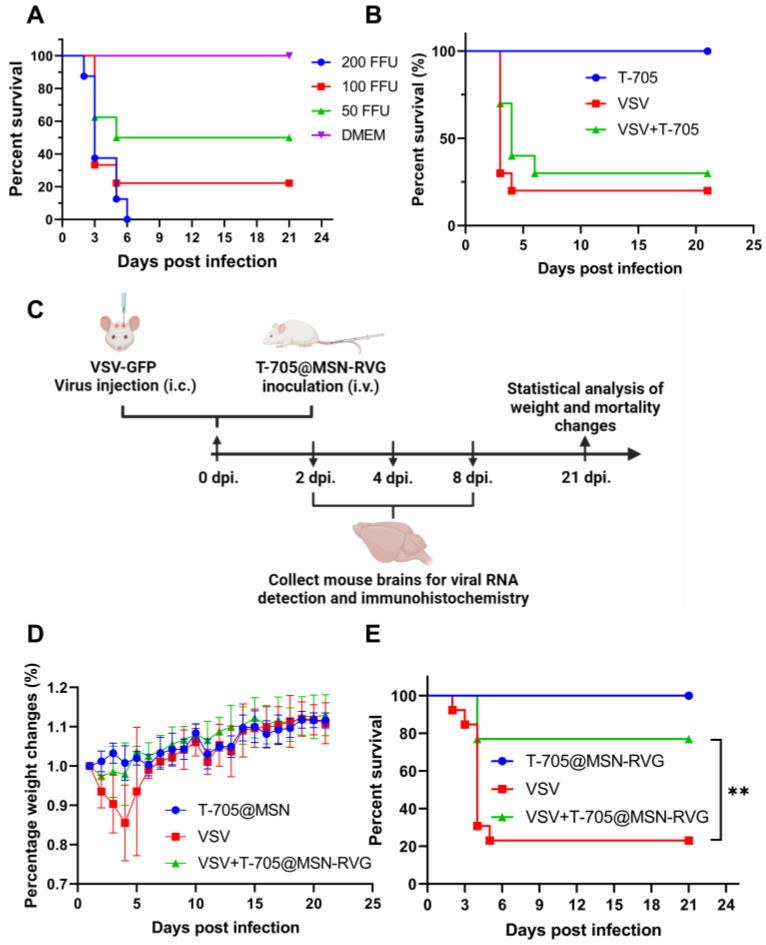
Antiviral treatment in mouse brains by T-705@MSN-RVG. (**A**) Six-week-old Balb/c mice were inoculated with VSV-GFP at 200 FFU, 100 FFU, 50 FFU, and DMEM by i.c. route (*n* = 10). (**B**) Six-week-old Balb/c mice (VSV-GFP infected at 100 FFU) were inoculated with 25 μL of T-705 (3 μg/mL) via i.v. route (*n* = 10). The mortality was recorded to analyze survival rates. (**C**) Schematic diagram of antiviral inhibition by using T-705@MSN-RVG. Six-week-old Balb/c mice (VSV-GFP infected at 100 FFU) were inoculated with 25 μL of T-705@MSN-RVG (2 mg/mL) via i.v. injection (*n* = 13). Bodyweight and mortality were recorded to investigate the (**D**) percentage weight changes and (**E**) survival rates. Statistical analysis of grouped comparisons was carried out by Log-rank (Mantel-Cox) test (** *p* < 0.01). The bar graph represents means ± SE.

**Figure 7 ijms-24-05851-f007:**
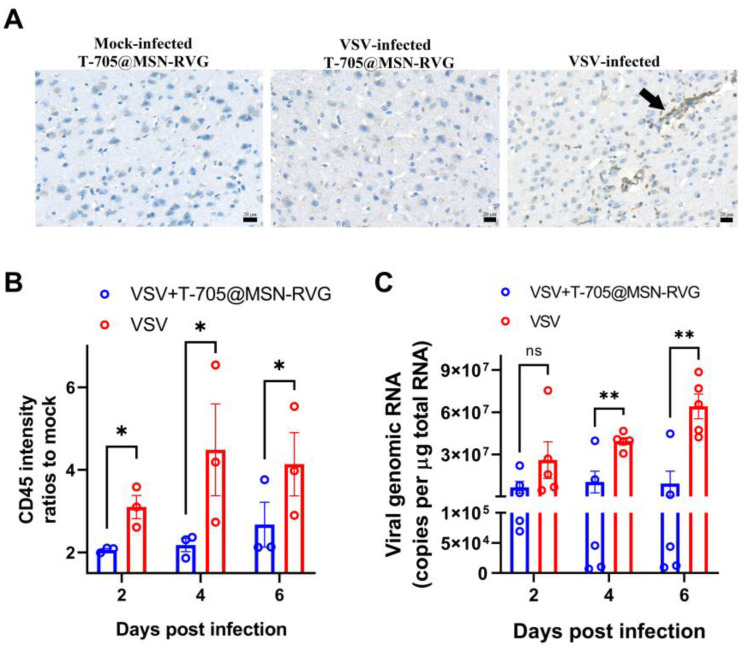
Changes in cell inflammation and viral load post treatment. Six-week-old Balb/c mice (VSV-GFP infected at 100 FFU) were inoculated with 25 μL of T-705@MSN-RVG (2 mg/mL) via i.v. route. (**A**) CD45 positive cells in mouse brains were visualized in the mock-infected T-705@MSN-RVG-inoculated group, VSV-infected T-705@MSN-RVG therapy group, and VSV-infected virus control group. Scale bar 20 μm. The black arrow indicates the CD45 positive cells with dark gray staining in the cells. (**B**) CD45 intensity ratios to mock and (**C**) viral genomic RNA (copies/μg total RNA) of the virus control group and VSV+T-705@MSN-RVG therapy group were analyzed at 2, 4, and 6 dpi. Statistical analysis of grouped comparisons was carried out by Student’s *t*-test (ns represents not significant, * *p* < 0.05; ** *p* < 0.01). The bar graph represents means ± SE.

## Data Availability

No new data were created or analyzed in this study. Data sharing is not applicable to this article.

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
