# Peer review of "RVG Peptide-Functionalized Favipiravir Nanoparticle Delivery System Facilitates Antiviral Therapy of Neurotropic Virus Infection in a Mouse Model"

_ijms, 2023, doi:10.3390/ijms24065851_

Round 1

Reviewer 1 Report

The study is well-designed and well-performed. The manuscript is also well-organized and well-written. There are no major points to be discussed.

Author Response

Reviewer #1:

The study is well-designed and well-performed. The manuscript is also well-organized and well-written. There are no major points to be discussed.

A: We are very appreciative of your kind comment about our work.

Reviewer 2 Report

Thank you for providing this nice study on preparing T-705@MSN-RVG  nanoparticle packaging favipiravir and  employing for antiviral treatment  in VSV-infected mouse brains. There are some issues that should be fixed before acceptance:

1. please provide us with more and more data related to physical and chemical properties of your MSN. 

2. Please check the cumulative release in a longer period. 

3. Please rewrite the conclusion part to make it as complete as to represent your aims and the impact of your study. 

Author Response

Reviewer #2:

Thank you for providing this nice study on preparing T-705@MSN-RVG nanoparticle packaging favipiravir and employing for antiviral treatment in VSV-infected mouse brains. There are some issues that should be fixed before acceptance:

  1. please provide us with more and more data related to physical and chemical properties of your MSN.

A: Thanks for the suggestions! The data of transmission electron microscopy (Figure 2E), elemental mapping (Figure 2F), particle size (Figures 1A, 1B, and, 1C), zeta potential (Figure 1D, 1E, and 1F), UV spectrum (Figure 2A), fluorescence (Figure 2B) were displayed in the revised manuscript to provide the physical and chemical properties of T-705@MSN-RVG product.

  1. Please check the cumulative release in a longer period.

A: We would like to thank the reviewer for the suggestion. We have extended the time point to 192 h in drug release experiment and updated the data in figure 2D in the revised manuscript.

  1. Please rewrite the conclusion part to make it as complete as to represent your aims and the impact of your study.

A: Thanks for the suggestion. We have rewritten the conclusion part to better represent the aim and impact of the study.

Reviewer 3 Report

The current manuscript presented by Ren et al. reported a novel nanoparticle delivery system that could potentially improve the treatment of neurotropic virus infections. The scope of this work fits the journal and the topic of the special issue well. The overall quality of this work is good but a few issues may need to be addressed before my final decision is made. These issues are listed below:

Major

1. Section 2.3, Page 3, line 104: I would like to know the reason/purpose why the authors included RABV here in the experiment? It seems VSV itself is enough for demonstration and all other sections are using VSV for demonstration.

2. Figure 2(E), Page 4, line 112: I notice that VSV-GFP infected cells imaged after 48 hours were gathered in a specific area (or maybe only these cells get infected). I would like to know why there is no diffusion of VSV-GFP particles during 48hr incubation. How the authors achieved this?

3. Section 2.6 and Figure 5A, Page 7&8: What does peak area mean or stand for in figure 5A? What is the role of this figure? I do not quite understand why the authors need the standard curve when studying drug release profile. Please provide additional explanations to clarify the purpose.

4. Section 3, Page 13, line 292-294: The authors claimed that the virus RNA copy numbers were reduced at 2, 4, and 6 dpi after mice were inoculated with nanoparticles. However, according to figure 7C, it seems that there is no significant decrease in virus RNA copy numbers. Personally, it looks like the virus copy numbers are maintained at a similar level, which may indicate that there is no new RNA being generated and the proliferation of the virus is paused. Please correct me if I read figure 7C wrong here. If not, then please explain why the authors believe the virus RNA numbers are decreasing.

5. Section 2.9 and last paragraph in discussion, Page 11-12&14: This paragraph of discussion along with section 2.9 is not very helpful. A few questions need to be addressed if the authors would still like to keep these parts in the article: 1. Is K371 site located at motif F in the VSV? 2. Any experimental/simulation evidence from the authors or other research groups can confirm the mutation at K371R in VSV will lead to attenuating inhibition efficiency? If not, then the authors’ inference may extend too far based on current results and knowledge. Simply observing protein model changes cannot be treated as solid evidence for author's discussion here in the last paragraph.

Minor

1. The authors may want to double check the use of abbreviations because they did not provide the full name for some abbreviations when they were first used in the article. For example, MSN on Page 2, line 73.

2. Technically speaking, the content in section 2.1 should not be a result but I understand the necessity of this paragraph. Personally, I suggest the authors could rewrite and fit it with last paragraph in introduction section (combine it with "Herein a T-705@MSN-RVG....mouse model."); or combine it with sections 2.5-2.7, which are talking about the characterization of fabricated nanoparticles, meaning the authors need to rearrange the order for sections 2.5-2.7 and move them to the beginning of results section. If the authors insist to keep it the same, I am okay with it too.

3. Page 2, line 84-85: I appreciate the authors' carefulness in the control group design. However, since the readers do not know the ratio of DMSO in the T-705 solution, the volume of added T-705 solution per well, and the volume of culture medium per well, I do not think the current info can tell readers the ultimate concentration of DMSO in each case, which makes it hard to figure out whether DMSO used in this experiment will harm cells or not. I suggest the authors could provide the estimated ultimate concentration of DMSO in each well, so the readers can compare this provided number with DMSO control group results.

4. Figure 1(B)(C)(D)(E) and Figure 2(C)(D): I personally recommend the authors to merge Fig 1B&1C into one figure, merge Fig 1D&1E into another figure, and merge figure 2C&2E into one figure. There is no meaning to separating them since they are showing the same thing. The current method to present these figures is confusing.

5. Figure 3B caption, Page 6, line 144: If the authors used Hoechst, then please just write about Hoechst. DAPI and Hoechst are not the same things. This type of expression may give wrong information to non-professional readers.

6. Page 7, line 169: I would like to confirm how the authors calculate 61.63% encapsulation rate based on Figure 5B? From my perspective, it seems that the drug release has not reached a plateau at 48hr in figure 5B. Also, it could be better if the authors could provide relevant numbers for the amount of drug packaged and the total amount of drug input here.

7. Figure 6(D)(E), Page 10, line 220-221: captions for (D) and (E) need to be swapped.

8. Figure 7(A), Page 11, line 233: Please provide a legend and indicate which color represents which. For example, which color indicates CD45 positive cells?

Author Response

Reviewer #3

The current manuscript presented by Ren et al. reported a novel nanoparticle delivery system that could potentially improve the treatment of neurotropic virus infections. The scope of this work fits the journal and the topic of the special issue well. The overall quality of this work is good but a few issues may need to be addressed before my final decision is made. These issues are listed below:

Major

  1. Section 2.3, Page 3, line 104: I would like to know the reason/purpose why the authors included RABV here in the experiment? It seems VSV itself is enough for demonstration and all other sections are using VSV for demonstration.

A: Thanks for the suggestions! Previously, we intended to use RABV as a viral control to demonstrate that favipiravir could also inhibit this virus. However, the VSV data was sufficient to support this study. So, we updated the Figure 4 by removing the data of RABV.

  1. Figure 2(E), Page 4, line 112: I notice that VSV-GFP infected cells imaged after 48 hours were gathered in a specific area (or maybe only these cells get infected). I would like to know why there is no diffusion of VSV-GFP particles during 48hr incubation. How the authors achieved this?

A: Thanks for the comments. The cells in this experiment were fixed in a growth medium containing 0.7% low-melting-point agarose and T-705. Since the cells were fixed by agarose, the viral particles can not be released into the supernatant (no supernatant, the medium is solidified by agarose) for rapid diffusion as in the liquid culture model. Under the MOI of 0.01, a small number of cells are infected. In this cell-fixed model, virus can just spread from infected cells to their neighbor cells. Therefore, the virus spot spread as a green circle in VSV control group. But the spot did not spread in the VSV+T-705 group, since the T-705 inhibited viral replication in the infected cells. The size of the fluorescence spot represented the extent of the virus spreading under different conditions. We have updated the result discrimination and its experimental section of Figure 4C to better describe its experimental design (section 4.6) and result (section 2.5).

  1. Section 2.6 and Figure 5A, Page 7&8: What does peak area mean or stand for in figure 5A? What is the role of this figure? I do not quite understand why the authors need the standard curve when studying drug release profile. Please provide additional explanations to clarify the purpose.

A: Thanks for the comments. The peak area represents the amount of drug content in the HPLC experiment. Because we need to calculate the amount of accumulative drug release. The drug amount needs to be calculated by using the value of the peak area and the equation of standard curve. So, we first obtain the mathematical relationship between the drug amount and the peak area through the standard curve equation (Figure 2C, section 2.2). The samples in drug release test were detected by HPLC to obtain their values of the peak area. By using the standard curve equation and the peak area, the drug concentration can be calculated to further investigate the cumulative drug release at different time point (Figure 2D, section 2.2). We have updated the result discrimination and its experimental section in section 2.2 and section 4.3.

  1. Section 3, Page 13, line 292-294: The authors claimed that the virus RNA copy numbers were reduced at 2, 4, and 6 dpi after mice were inoculated with nanoparticles. However, according to figure 7C, it seems that there is no significant decrease in virus RNA copy numbers. Personally, it looks like the virus copy numbers are maintained at a similar level, which may indicate that there is no new RNA being generated and the proliferation of the virus is paused. Please correct me if I read figure 7C wrong here. If not, then please explain why the authors believe the virus RNA numbers are decreasing.

A: Thanks for the comments. Based on survival rate experiments, not all mice survived (survival rate of 77%). The higher viral copy number in the brains of mice that were about to die after 6 days resulted in raising the mean viral copy number of the treated group (seen below, highlighted by green box). Thus, there was no decreasing trend in the mean RNA copy number of the mice on days 4 and 6 compared to their previous day (day 2), according to our data. Although the vRNA copy numbers in survived mice were reduced (Data of days 4 and 6 in Figure 7C, the small circles in the blue bar graph, seen below, highlighted by yellow Box) at days 4 and 6 compared with day 2. Please see the above mentioned Figure in the submitted word format file.

Therefore, we just intended to highlight the significant reduction in RNA copy number compared to the viral control group on days 4 and 6. We are not trying to emphasize the decreasing trend for the therapy group itself in the progress of days 2, 4, and 6. We have checked and revised the description in the text to avoid wrong description of this point (section 2.8 and discussion section).

  1. Section 2.9 and last paragraph in discussion, Page 11-12&14: This paragraph of discussion along with section 2.9 is not very helpful. A few questions need to be addressed if the authors would still like to keep these parts in the article: 1. Is K371 site located at motif F in the VSV? 2. Any experimental/simulation evidence from the authors or other research groups can confirm the mutation at K371R in VSV will lead to attenuating inhibition efficiency? If not, then the authors’ inference may extend too far based on current results and knowledge. Simply observing protein model changes cannot be treated as solid evidence for author's discussion here in the last paragraph.

A: Thanks for the kind suggestions! We agree that the figure of the protein model is not very helpful. We removed these models. We just showed the sequence-aligned data to prove that the lysine site is conserved in the RdRp of different neurotropic RNA viruses. Since it is a binding site for T-705 to enter the RdRp to inhibit RNA replication, its conservativeness in different viruses is helpful for virus inhibition by T-705. The sequence aligned data was merged to Figure 4D in the revised manuscript.

Minor

  1. The authors may want to double check the use of abbreviations because they did not provide the full name for some abbreviations when they were first used in the article. For example, MSN on Page 2, line 73.

A: Thanks for the kind reminding. We have checked and revised the abbreviations used first time in the manuscript.

  1. Technically speaking, the content in section 2.1 should not be a result but I understand the necessity of this paragraph. Personally, I suggest the authors could rewrite and fit it with last paragraph in introduction section (combine it with "Herein a T-705@MSN-RVG....mouse model."); or combine it with sections 2.5-2.7, which are talking about the characterization of fabricated nanoparticles, meaning the authors need to rearrange the order for sections 2.5-2.7 and move them to the beginning of results section. If the authors insist to keep it the same, I am okay with it too.

A: Thanks for the suggestion. We have removed the paragragh in original section 2.1 and placed the original Figure 1A (the flow chart of nanoparticle synthesis and application) as the Graphical Abstract to better introduce the aim and progress in this study. The original sections 2.5-2.7 were moved to the beginning at sections 2.1-2.3 as Figures 1 and 2 in the revised manuscript (section 2.1-2.3 in the revised manuscript).

  1. Page 2, line 84-85: I appreciate the authors' carefulness in the control group design. However, since the readers do not know the ratio of DMSO in the T-705 solution, the volume of added T-705 solution per well, and the volume of culture medium per well, I do not think the current info can tell readers the ultimate concentration of DMSO in each case, which makes it hard to figure out whether DMSO used in this experiment will harm cells or not. I suggest the authors could provide the estimated ultimate concentration of DMSO in each well, so the readers can compare this provided number with DMSO control group results.

A: Thanks for the suggestion. We have added the discrimination of DMSO concentration used in the viability test in experimental section (section 4.4) and result (section 2.4).

  1. Figure 1(B)(C)(D)(E) and Figure 2(C)(D): I personally recommend the authors to merge Fig 1B&1C into one figure, merge Fig 1D&1E into another figure, and merge figure 2C&2E into one figure. There is no meaning to separating them since they are showing the same thing. The current method to present these figures is confusing.

A: Thanks for the suggestion. We have merged the original figures 1B&1C, and figures 1D&1E to generate the merged figures 3A and 3B in the revised manuscript (section 2.4).

  1. Figure 3B caption, Page 6, line 144: If the authors used Hoechst, then please just write about Hoechst. DAPI and Hoechst are not the same things. This type of expression may give wrong information to non-professional readers..

A: Thanks for the kind reminding. The description of the figure caption has been revised in figure 5B (section 2.7) and kept the same with that in experimental section (section 4.7).

  1. Page 7, line 169: I would like to confirm how the authors calculate 61.63% encapsulation rate based on Figure 5B? From my perspective, it seems that the drug release has not reached a plateau at 48hr in figure 5B. Also, it could be better if the authors could provide relevant numbers for the amount of drug packaged and the total amount of drug input here.

A: Thanks for the kind review. We provided the calculation method and drug amount in the result section (section 2.2) and experimental section (section 4.3).

  1. Figure 6(D)(E), Page 10, line 220-221: captions for (D) and (E) need to be swapped.

A: Thanks for the suggestion. The Figure 6D and 6E have been swapped in the updated Figure 6 (section 2.8).

  1. Figure 7(A), Page 11, line 233: Please provide a legend and indicate which color represents which. For example, which color indicates CD45 positive cells?

A: Thanks for the suggestion. The pointing arrow was added in Figure 7A to indicate the CD45 positive cells (section 2.8).

Round 2

Reviewer 3 Report

The authors have addressed all my comments sufficiently. I have no further comments.